# An Energy-Efficient Fail Recovery Routing in TDMA MAC Protocol-Based Wireless Sensor Network

**Odilbek Urmonov and HyungWon Kim \***

Electronics Engineering Department, School of Electronics Engineering, Chungbuk National University, Cheongju 371763, Korea; odilbek@cbnu.ac.kr
**\*** Correspondence: hwkim@cbnu.ac.kr; Tel.: +82-43-261-2399

**Abstract:** Conventional IoT applications rely on seamless data collection from the distributed sensor nodes of Wireless Sensor Networks (WSNs). The energy supplied to the sensor node is limited and it depletes after each cycle of data collection. Therefore, data flow from the network to the base station may cease at any time due to the nodes with a dead battery. A replacement of the battery in WSNs is often challenging and requires additional efforts. To ensure the robust operation of WSNs, many fault recovery routing mechanisms have been proposed. Most of the previous fault recovery routing methods incur considerable delays in recovery and high overhead in either energy consumption or device cost. We propose an energy-efficient fail recovery routing method that is aimed to operate over a data aggregation network topology using a TDMA media access control (MAC). This paper introduces a novel fault recovery routing algorithm for TDMA-based WSNs. It finds an optimal neighbor backup parent (NBP) for each node in a way that reduces the energy consumption. The proposed method allows the NBPs to utilize the time slot of the faulty parent nodes, so it eliminates the overhead of TDMA rescheduling for NBPs. To evaluate the fault recovery performance and energy efficiency of the proposed method, we implemented it in C++ simulation program. Simulation experiments with an extensive set of network examples demonstrate that the proposed method can extend the network lifetime by 21% and reduce the energy consumption by 23% compared with the reference methods.

**Keywords:** failure recovery; routing; wireless sensor network; redundant path; backup parent node; network lifetime and power consumption; active period

## 1. Introduction

The recent advancement of WSNs has enabled a variety of Internet of Things (IoT) applications that penetrate our daily life [1]. Many IoT applications are often safety related and mission-critical (e.g., health care, active volcano monitoring, fire alert, etc.) where device failures might cause serious consequences [2,3]. Especially, wireless sensor nodes deployed for environment monitoring, periodically send their sensing data to a gateway called a sink node in a multi-hop topology [2].

Sensor nodes are used widely in the industry to monitor and accumulate the data related to the object. For instance, deploying sensor nodes, we can receive periodic information about the environments such as wild nature (forests or deserts), special industrial facilities, etc. [2]. We may apply WSN to obtain up-to-date temperature information or monitor toxic gas levels in different branches of industry. Large-scale self-organized wireless sensor and mesh network provide an opportunity to develop Smart Environment and Smart Grids applications [1]. The WSN is critically important to support these advanced applications.

In the past, many WSNs employed a carrier sense multiple access (CSMA) protocol due to its simplicity [4]. Such networks, however, share the media and therefore suffer from frequent

collisions, which incur retransmissions of packets causing extra energy loss. A time division multiple access (TDMA) protocol is regarded as an effective alternative to CSMA, since it can ensure fair and collision-free data forwarding from all nodes, therefore reducing the energy loss [4,5]. Our proposed method is thus based on TDMA. Regardless of the choice of protocol, however, any WSN is susceptible to devise failure or battery depletion, and therefore it may lose network connectivity.

Recent studies on WSNs have achieved considerable enhancement in network architecture and data forwarding protocols to reduce the energy consumption [2]. The primary goal of many WSNs is to maximize the network lifetime even under the event of node failures [6]. Hence, it needs a fail recovery method that operates the rest of the WSN to maintain the desired lifetime.

For low-power WSNs, a tree structure topology is often adopted [7], since it permits simple routing paths from all the nodes towards the sink (root) node, which acts as a gateway collecting all the sensing data. In WSNs of tree structure topology, each child node at a lower level forwards its sensing data to its parent node at a higher level until all data are delivered to the sink node [7]. If any parent node fails, then, all nodes in the subtree under the failed parent lose their routing path towards the sink node. A large portion of the network, therefore, can be isolated, resulting in all their sensing data being lost. Figure 1 illustrates such a faulty parent and its isolated subtree marked by a dotted line.

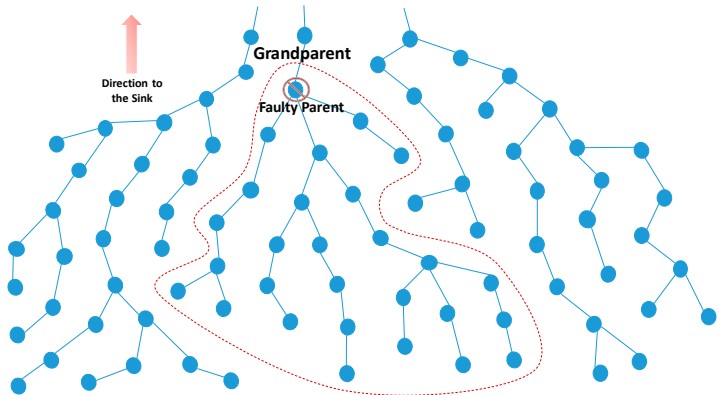

**Figure 1.** An example of isolation in WSN due to single parent fault.

There are many causes of node failures such as sensor hardware impairment, radio frequency (RF) transceiver malfunctioning, and battery depletion [8]. In the field of network fail recovery, many previous researchers consider battery depletion as the most common cause of node failures [8–10]. Our work also assumes battery depletion as the cause of node failures for the sake of presentation, while the proposed recovery algorithm can be extended to any types of node failures.

When the number of faulty nodes exceeds a certain level, the network may cease to operate. The time until the first sensor node runs out of energy is called the First-node Die-Time (FDT). The period from FDT to the time when all the sensor nodes are dead, or the network is completely disabled, is called All-node Die-Time (ADT) [11,12].

As the percentage of faulty nodes in the network exceeds the threshold of the fault ratio, the network is considered as disabled and the remaining alive nodes of the network become useless. The network lifetime is defined as the duration from a network initialization to the time when the network is disabled [13]. Our goal is to restore the connection between isolated nodes and the functioning portion of the network.

Many studies have shown that the occurrence of faults in WSN is largely classified into two groups: (i) transmission fault and (ii) node fault. The node fault is further classified into five categories: power fault (battery depletion), sensor circuit fault, microcontroller fault, transmitter circuit fault, and receiver circuit fault [10,14]. In the cases of receiver or transmitter fault, the sensor node cannot receive nor send its sensed data as well as the data forwarded from the child nodes. The sensor circuit failure is considered as less critical, as the sensor node can still forward the data from its child nodes [15].

Depending on the hardware condition of the sensor node, they are categorized as a Normal node, Traffic node, End node, and Dead node [10]. According to the results of previous studies, the categorization helps in reducing the percentage of dead nodes in the network, therefore improving the network lifetime.

As electronic components of wireless sensor nodes are becoming more reliable, battery depletion is considered as the most prominent source of failure according to recent reports [16]. For the WSNs in a harsh environment, a distributed fault detection (DFD) algorithm was proposed by Reference [17]. The DFD algorithm does not incur additional transmission costs because they use existing network traffic to identify sensor failures. Due to the exchange of multiple enquiry messages, this method may consume more energy during the recovery process. Another common solution to providing fault tolerance (FT) is adding redundant hardware or software [18]. Highly stringent design constraints (e.g., limited battery power) of WSNs, however, make it difficult to add such redundancy due to the additional cost.

In [19], the authors proposed a method of fault recovery during the routing process in a WSN. It classifies the fault recovery methods into two main classes based on the improvement in data transmission. The first technique is retransmission, in which the source retransmits the data through another path when the original path fails. The second technique is data replication which duplicates the data to multiple copies over multiple paths. Utilization of multiple paths for the same message may reduce the network efficiency and cause additional contention to the channel access.

In [20], it is studied a temporal classification method that classifies fail recovery techniques as preventive and curative. Preventive techniques attempt to keep the network functioning without any interruption when any fault occurs. In contrast, curative techniques employ a reactive process that interrupts the network functions while it recovers an identified fault.

The methods of [21] are examples of preventive techniques. They select in advance the second-best routing option as a redundant path to use when a fault appears in the shortest path. To meet the energy efficiency requirement, their algorithm utilizes the largest portion of the shortest path that can still forward the data to determine the redundant path. Since the nodes within the shortest path execute multiple transmissions, they may consume greater energy than the other nodes. An unequal distribution of network load may cause a failure of nodes in the shortest paths. Then, the system frequently executes a fail recovery procedure which makes the nodes consume additional energy.

The authors of [22] proposed a routing protocol that allows real-time fault recovery. It uses the remaining time of each packet and the state of the forwarding candidate set of nodes, and chooses a path dynamically. Upon detection of a failure, sensor nodes change their status to the jump mode and dynamically adjust the probability of jump to increase the ratio of successful transmissions. Updating the state of the nodes requires additional control message exchange which can be costly in the network with limited energy.

In [23,24], it is reported meta-heuristic fault detection algorithms to overcome WSN failure and improve the system reliability. Like many previous approaches, however, such fault recovery methods add significant overheads to both hardware and power, and thus are unacceptable for practical IoT networks.

In [16], S. Gobriel et al. recommended classifying the edges between sensor nodes into three types: primary, backup and side edges. Each node selects one parent as a primary parent and zero or more parents as backups. Primary edges from a spanning tree are used as long as no communication error occurs. If an error occurs in a primary edge, data may be successfully delivered by one of the backup edges. Authors of this work, however, did not clearly specify on what basis their algorithm selects the primary and backup edges.

In this paper, we propose a fault recovery routing algorithm called an energy efficient, neighbor-extended maximal connectivity re-routing (NE-MCR), which does not incur any additional hardware cost, and thus is well suited to WSNs under stringent power constraints. The NE-MCR algorithm conducts an additional route-recovery process after the main routing and TDMA scheduling

steps are completed. In this, we identify as faulty nodes the parent nodes that do not respond (acknowledge) to their child nodes within a given time duration. Our method selects the local optimal backup parent nodes during the routing process in a way that ensures most of the child nodes can maintain their connectivity.

In WSNs, a number of different techniques can be applied to detect the failures. Some researchers proposed the use of passive information collection for the purpose of failure detection. In these methods, information crucial to detect the failure can be extracted from regular data packets sent to the sink node [25,26]. In [8], the authors proposed a special framework to detect the failure in WSN. In this model, sensor nodes piggyback checksum tags of path upon all regular messages sent to the sink node. Each node updates the tags with its own node identification (ID) by means of the Fletcher checksum algorithm. After receiving packets from all routes, the sink node inspects their checksum. To identify any changes in a specific path, the sink node injects a series of control messages. Based on the response to these messages, the sink determines and reports the failure. Most failure detection algorithms use additional control information which incurs an overhead in low power WSN. Therefore, in the current work, faulty parents are detected by identifying the nodes that do not acknowledge their child within a given period.

We compare the NE-MCR algorithm with two reference algorithms, i.e., exponential and sine cost function-based routing (ESCFR) and double cost function-based routing (DCFR) that are presented in [11]. Our simulation results prove that NE-MCR has improved energy efficiency and a longer network lifetime than reference algorithms.

We are targeting the applications of the proposed method's IoT networks for wireless metering. In this application, every sensor node periodically wakes up at the same time and sends its sensing data towards the sink node via pre-calculated multi-hop routing paths. We consider a contention-free TDMA protocol, where each node transmits at its time slot, which is pre-allocated during the scheduling process after the routing process is done [27]. The essence of this network structure can be explained in a way that each parent node receives all its child nodes' data and aggregates them into one data packet along with its own sensing data. It then transmits its aggregated data to its parent node at its allocated time slot. To conserve the battery power, each parent node switches back to sleep mode once it transmits the aggregated data. We also implemented above procedures on real hardware and demonstrated data forwarding performance in [28]. The authors of this paper believe that there is no related research proposing an energy-efficient fail recovery routing algorithm for this specific type of network model.

In most environmental monitoring, facility diagnosis and wireless metering applications employing energy-harvesting devices like solar cells are not effective solutions since sensor devices are usually installed in an indoor environment or the dark basements of buildings. Using a larger battery is not acceptable due to the cost and size constraints on the sensor devices, since these applications are often deployed throughout the entire city to monitor temperature, air pollution or toxic gas level.

The remainder of this paper is organized as follows. In Section 2, we introduce our network topology and energy model for tree-structured WSN. Section 3 elaborates the proposed fault recovery re-routing algorithm. The definition of NE-MCR algorithm's cost (objective) function is explained in Section 4. Performance evaluation of the NE-MCR algorithm is provided in Section 5, followed by the conclusion in Section 6.

## 2. Network Topology and Energy Model

### 2.1. Network Topology with Time Division Multiplexing

This section describes the network topology, scheduling and routing schemes of the proposed method. Energy-efficient data aggregation is often considered as the primary goal of many WSNs. In conventional sensor networks based on simplistic CSMA, as each sensor's data travels through multi-hop paths, its data is duplicated and transmitted by the nodes along the paths. Such duplicate

transmissions, however, often cause excessive energy consumption. In this paper, we consider a more energy-efficient data forwarding method, a TDMA-based aggregate-and-forward method with convergent network topology. In this forwarding method, each node receives sensing data packets from all its child nodes in different time slots, and sends at once an aggregated data packet to its parent node in another time slot. In the network topology considered in this paper, we assume that all the nodes wake up together at a pre-scheduled sensing period, while they stay in a long sleep period in order to save energy.

In [27], the authors proposed a multi-channel TDMA scheduling where each sensor node has a single radio and selects one channel from a set of RF channels. We also consider a similar TDMA scheduling method in this paper. The scheduling process is conducted after the routing process. In the routing process, each child node selects its parent considering the transmission distance. In the scheduling process, each selects a time slot while satisfying the constraint that the time slot of a parent node $p_i$ is higher than all child nodes $c_i$. This allows all nodes to aggregate and forward the sensing data from leaf nodes towards the final destination, the sink node. For example, Figure 2a illustrates a network with TDMA time slots and channels selected by the above routing and scheduling process for each node. This example is cited from [27]. In this network scenario, a sensor node first selects its parent node, and then schedules its transmission in the time slot that is earlier than the slot of the selected parent. Employing different RF channels allows concurrent time slots for large-scale networks. It also mitigates the interference between the nodes that are using identical slots in the zone of interference. In Figure 2, node n31 and n33 select the same time slot. Since they use different channels, their concurrent transmission does not cause collision. Although each sensor node has a single radio to communicate, it can bridge the child nodes that use various RF channels with the sink node. Initially, it tunes the RF channel of the child node who is allotted with the earliest slot. Then, it switches to other channels according to the sequence of the slots assigned to child nodes. The time consumed for switching from one channel to another is negligible [27]. For instance, in Figure 2, node n11 receives data from the child nodes using channel 1 and 2. Then, it forwards the aggregated data to the sink node. The process of data forwarding in various time slots is depicted in Figure 2b.

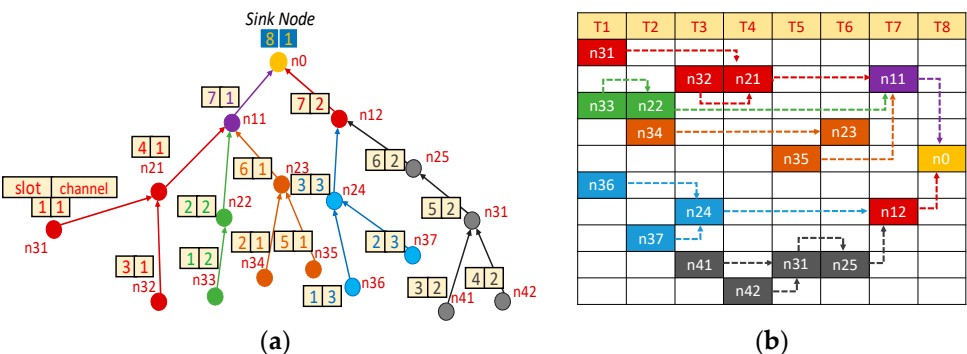

**Figure 2.** Example, TDMA based data forwarding in WSN: (**a**) presents the data flow in the network; (**b**) illustrates the TDMA scheduling table.

The scheduling process ensures another constraint that multiple nodes can share the same time slot only when their channel is different. Figure 2b shows the result of time-slot scheduling. It uses eight time-slots and three channels to complete the aggregate and forward process from all nodes to the sink node n0. Shaded boxes denote each node-allocated time slot, while the colors of the boxes indicate the different channels selected. The dotted arrows represent the forwarding path from a child to its parent node.

The objective of the routing and scheduling algorithm is to minimize the energy consumption by minimizing the number of time slots without exceeding the specified number of channels. If the network forwards all data using fewer time slots, all its nodes can switch back to sleep mode early, leading to less active energy. The authors of [27] proposed a binary linear programming method to

solve the time slot allocation problem and implemented a heuristic algorithm distributed method in each node. Since, the method uses in [27] is a more practical method that can be applied to a network of large scale, we have employed it as the routing and scheduling method.

The resulting network topology of the above routing [5] and scheduling process [27] restricts every node to have only a single egress edge while allowing multiple ingress edges. When the WSN wakes up, all nodes wake up at the same time and measure their sensing data at the same time. Then each node starting from the leaf nodes forwards its data to its parent node. Each parent node waits until all sensing data are received from its child nodes. Then, the parent node aggregates all the received sensing data into one data packet and forwards the aggregated data to its parent node in the next hop. Our system model uses the routing algorithm in [5] and the scheduling algorithm in [27] during the network initialization stage.

### 2.2. Energy Model of Convergent Network

This section presents an energy model of the proposed convergent networks. Figure 3 shows a subtree of six nodes, where parent node 3 has five child nodes. All five child nodes must be allocated in different time slots since they cannot transmit to the same parent node in the same time slot. During the five time slots, node 3 receives data from its all child nodes, and consumes reception energy for each ingress (child) node. The sum of reception energy $E_A^{ingress}$ of all ingress nodes for a parent node p is expressed by Equation (1).

$$E_A^{ingress} = \sum_{i=1}^{n} E_{rx}^i, \tag{1}$$

Here, $n$ denotes the number of child nodes, which is five, as shown in Figure 3. We assume, that the energy consumed by sensing and data processing is negligible [29] in order to focus on the problem of minimizing the data forwarding energy—a primary cause of energy consumption.

In Equation (1), the reception energy $E_{rx}^i$ for the received data from node $i$ to node $p$ is given by Equation (2) [30].

$$E_{rx}^i = l_i \cdot P_{elec} \tag{2}$$

Here, $P_{elec}$ is the power consumed by the transceiver and radio circuit including the channel coding and modulation circuits. $l_i$ indicates the data length in seconds.

The total energy consumed by each node during one active period of sensing and data forwarding is estimated by Equation (3):

$$E_A^{Total} = \sum P_{rx} T_{rx} + \sum P_p T_p + P_{tx} T_{tx} \tag{3}$$

Here, $E_A^{Total}$ is the total energy consumed by a node during its active period; $T_{rx}$, $T_p$, and $T_{tx}$ denotes the time spent on receiving, processing and transmitting respectively (sum of these time segments equal to active period), and $P_{rx}$, $P_p$, and $P_{tx}$ indicate the amount of power consumed by receiving, processing and transmission operations, respectively. Equation (3) can also be expressed by Equation (4) assuming $E_P^{Total}$ is the energy expense during the active period of the node.

$$E_A^{Total} = \sum_{i=1}^{n} E_{rx}^i + \sum_{i=1}^{n} E_p^i + E_{tx} \tag{4}$$

Here, $E_{tx}$, $E_{rx}$, and $E_p$ denote transmission, receiving and processing energy, respectively, while $n$ indicates the number of child nodes of the current node.

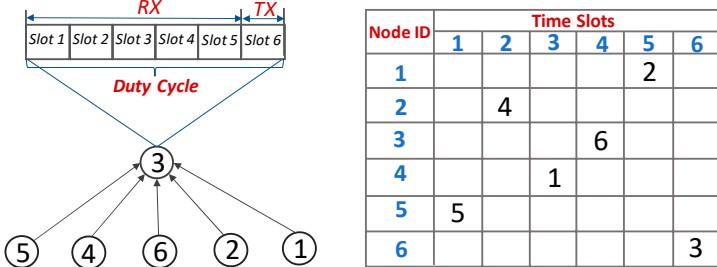

**Figure 3.** Power consumption of parent node in duty cycle.

In multi-hop sensor networks, the transmission power is often constrained. If the transmission power is increased beyond that constraint, it may cause interference with other nodes. The transmission distance and packet length are the main arguments for the transmission energy function. These two parameters are proportional to the energy consumption. Let's suppose that initially node-A's transmission power is set approximately to 5 dBm and it is aimed to cover a 150 m range with 95% packet delivery ratio (PDR). Node-A may create a 200 m round interference zone for other neighbor nodes that use the same slot. Hence, these nodes should not execute transmission when node-A sends its data to the parent. However, nodes within 250 m are allowed to transmit data using the same slot since their transmission is not interfered with by node-A. If node-A increases its transmission power to 10 dBm, then its interference zone obviously enlarges, and it interferes with the nodes within a 250 m range. On the other hand, now node-A consumes twice as much energy for each transmission and its battery suffers from intense drainage of energy. Thus, in strictly-scheduled TDMA MAC protocol-based WSNs, we cannot merely increase the power of transmission due to the above constraints; we can, however, reduce the transmission power when the failure recovery process replaces a failed node by a backup node in a way that the transmission distance is reduced [31]. Thus, we assume that every node is assigned a constrained transmission power, which leads to the same maximum transmission energy consumption $E_{tx}$ for every node for its allocated time slot. Although the receiving and processing energy may increase during the fault recovery (backup node selection) procedure, we assume the processing power is negligible compared with the receive and transmit power [32]. On the other hand, an increase in selected backup parent node's receiving energy is inevitable since its number of child nodes grows due to the failure recovery process. When a sensor node is selected as a backup node that has a larger number of child nodes, it consumes higher energy to receive the additional sensed data from the additional child nodes.

For the calculation of a network lifetime, we calculate each sensor node's battery lifetime $L$. The lifetime $L$ of a node is defined as the time length from the power-up time until the battery outage time of the node, which is expressed by Equation (5).

$$L = \frac{E_{initial}}{E_{consumed}} \times t_{cycle} \tag{5}$$

Here, $E_{initial}$ is the initial energy of node $n_i$ at its power-up time. $t_{cycle}$ is one full cycle time including data sensing, receiving all child data, and transmitting the aggregated data to $n_i$'s parent node. $E_{consumed}$ denotes the energy consumed by the node $n_i$ during $t_{cycle}$. $E_{consumed}$ is expressed by Equation (6) ignoring the energy consumption for sensing and data processing.

$$E_{consumed} = E_{tx}(l, d) + n E_{rx}(l) \tag{6}$$

In Equation (6), $n$ denotes the number of child nodes and the transmission power $E_{tx}(l, d)$ for data length $l$, and distance $d$ is defined as follows. Using Frii's path loss model, we assume that the transmission power of node $n_i$ is selected in a way that the path loss is compensated [33]. According to

the radio model used in Reference [34], data transmission usually depends on the distance and packet length, as expressed in Equation (7).

$$E_{Tx}(l,d) = \begin{cases} l \cdot E_{elec} + l \cdot E_{Fs}d^2, & if\ d < d_0 \\ l \cdot E_{elec} + l \cdot E_{amp}d^4, & if\ d > d_0 \end{cases} \tag{7}$$

Here, $E_{Fs}$ and $E_{amp}$ are the amplifier energy consumption for the distances in free space ($d^2$ power loss) and the channel with multi-path fading ($d^4$ power loss) respectively. It is mentioned above that a variable $l$ is a length of data in Equation (7) (or it denotes a time that is required to send a sensed data). Since we are concerned with finding backup nodes with a short distance, the free space fading ($E_{Fs} \times d^2$) is a more appropriate model [35], and thus, it is used in the remainder of the paper.

Using Equations (6) and (7), Equation (5) can be expressed by Equation (8).

$$L = \frac{E_{initial}}{E_{tx} + nE_{rx}} \times t_{cycle} = \frac{E_{initial}\ t_{cycle}}{lE_{elec} + lE_{Fs}d^2 + nlE_{elec}} = \frac{E_{initial}\ t_{cycle}}{lE_{Fs}d^2 + (n+1)lE_{elec}} \tag{8}$$

Equation (8) indicates that the distance and the number of child nodes are the main components of energy consumption.

Using Equation (8), the proposed algorithm selects a set of optimal backup nodes for all parent nodes in the network as described in the next section.

## 3. Proposed Route-Recovery Algorithms

### 3.1. Energy Model of Convergent Network

A single sensor node failure can cause branch isolation and thus leave many nodes with broken routes to the sink node. To recover the connectivity, we propose a route-recovery method called maximum connectivity local rerouting (MCR). It quickly replaces the faulty parent node with a backup parent among the child nodes, and thus restricts the rerouting process to only the local nodes within one hop subtree of the faulty node. For example, consider an example network in Figure 4a, where a parent node p is faulty. Here, MCR selects $c_k$ among p's child nodes as a backup parent $MCR\_BP(p)$. For fast recovery in the event of a fault, MCR operates in two stages. The first stage is a processing step for backup parent selection, which is conducted as a part of the proactive routing algorithm. The second stage is a recovery step during which a real-time rerouting operation is conducted only when a fault occurs. The recovery step instantly replaces the faulty node by the pre-selected backup parent node. Hence it does not interrupt the forwarding operations of all other nodes.

The first stage of MCR, the preprocessing algorithm, is executed during the network initialization period. It examines every child node $c_i$ of each parent node p for two connectivity conditions: (1) How many sibling nodes of $c_i$ are covered by $c_i$'s wireless range; and (2) whether $c_i$ can reach its grandparent $g$ (the parent of $p$). This method selects one of the child nodes who best satisfies the above two conditions as a backup parent for its primary parent. If $p$ fails in the future, $c_i$ instantly replaces $p$'s role. In other words, $c_i$ receives the data from $p$'s child nodes and forwards the aggregated data to the grandparent $g$. The key advantage of this recovery process is that the selected $MCR\_BP$ $c_i$ inherits the time slot of $p$, which eliminates the need for time slot rescheduling of many nodes around $p$. For example, consider the example subtree of Figure 3. Suppose that the MCR algorithm's preprocessing step selected node 3 as the backup $MCR\_BP$; If the parent node 6 fails, node 3 takes over the parent's role of node 6 and the time slot Slot6. Once node 3 receives and aggregates other sibling's data, it then transmits the aggregate data to the grandparent node using Slot6. The proposed algorithm, therefore, recovers the communication failures without disturbing the surrounding nodes except the siblings of $c_i$. In contrast, most of the previous fail recovery algorithms either re-allocate the parent or reroute the orphan child (the node who lost a primary parent) nodes to another parents in the neighborhood, thus disrupting many neighbor nodes.

The MCR algorithm's preprocessing procedure for the backup parent selection is illustrated in Figure 4b. It searches for the best candidate of a backup parent for every parent node $p_j$ amongst $p_j$'s child nodes to prepare for the event when $p_j$ ever fails.

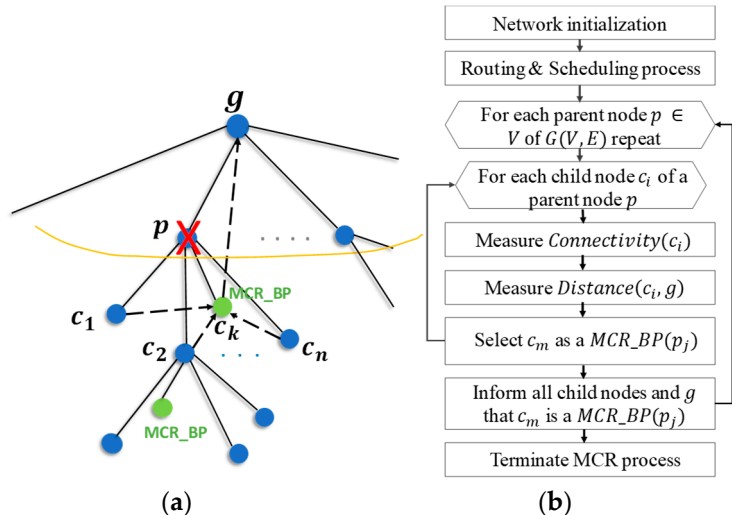

**Figure 4.** MCR back-up parent selection procedure. (**a**) Segment of a network where the primary parent failed; (**b**) $MCR\_BP$ selection procedure.

Procedure Select-MCR_BP

Let's assume $G(V, E)$ represents our network [36], where $V$ is a set of vertices (nodes) and $E$ is the number of edges (lines).

For each parent node, $p_j \in V$ of $G(V, E)$ repeats the following steps:

- For each child node $c_i$ of a parent node $p_j$:

  (1)    Measure $Connectivity(c_i)$ = number of other child nodes $c_k$ of $p_j$ such that all $c_k$ are in the wireless range $W_i$ of $c_i$.

  (2)    Measure $Distance(c_i, g)$ = the distance from $c_i$ to grandparent $g$ of $c_i$ (parent of $p$).

- Select $c_m$ as a MCR backup parent $MCR_{BP}$ of $p_j$ such that $Connectivity(c_m)$ is maximum and $Distance(c_m) < W_m$, where $W_m$ is the wireless range of $c_m$.
- If for all child nodes, $c_i$ $g$ is unreachable, select the $c_m$ as a $MCR\_BP(p_j)$ and NE-MCR concept to find NE-$MCR\_BP(p_j)$ (details are given in following subsection).
- Inform all $c_k$'s and g that $c_i$ is chosen as $MCR\_BP(p_j)$.

When $MCR\_BP(p_j)$ is selected for each $p_j$, $p_j$ informs its child nodes $c_k$ and grandparent node $g$ by broadcasting a message $M(MCR\_BP(p_j), slot(p_j))$. Then nodes $c_k$ and $g$ record the node ID of $MCR\_BP(p_j)$ and the time slot of slot$(p_j)$. This process completes the procedure of $MCR\_BP$ selection. Then, during the main data-forwarding operation, if a fault occurs in node $p_j$, the real-time recovery process is conducted as follows: All child nodes $c_k$ of $p_j$ forward their data to pre-selected $MCR\_BP(p_j)$ instead of the failed parent $p_j$. Then, $MCR\_BP(p_j)$ forwards its data to the grandparent $g$, while bypassing the failed node $p_j$. The failed recovery process takes place only when a fail occurs in the parent node whose backup node was pre-selected. We have implemented an MCR simulator in a C program and measured the performance of MCR using an example network of size 1000 nodes. We evaluated the behavior of the number of isolated nodes (nodes with lost routing) by injecting faults to an increasing number of nodes. Figure 5 compares the number of isolated nodes for the two-fail recovery method. In the case of the MCR fail recovery method, the number of isolated nodes tends to grow linearly, whereas in a naïve route recovery method (based on random selection),

the number of isolated nodes grows exponentially. The significant reduction in the number of isolated nodes is attributed to the fact that MCR can efficiently select optimal backup nodes, whereas the random selection method could not find proper backup parents in many cases. The proposed recovery algorithm selects the node that has maximal connectivity with other siblings. In the sparse network scenario, however, the elected $MCR\_BP(p_j)$ may not cover all siblings. Therefore, in Figure 5, our method experiences additional isolated nodes when the number of induced faulty nodes grows.

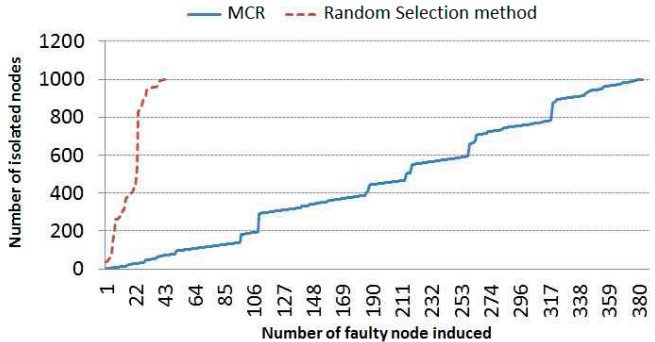

**Figure 5.** Comparison of the number of isolated nodes for the WSN of 1000 nodes when the number of faults is increased.

In the event of a parent node's failure, the first action of its pre-selected backup parent (BP) is to take over the faulty parent's time slot. The selected BP, then, notifies all its sibling nodes that BP is selected to act as a backup parent. This pre-processing method is carried out only once when the network is initiated, and its initial routing is conducted. A detailed pseudo code for the backup parent selection algorithm is given in [37].

The MCR is a fast algorithm with low complexity and no hardware overhead. It may, however, fail in finding a BP node when none of the child nodes can reach its grandparent node. We define this problem as an out-of-reach problem. Increasing the wireless range of the child nodes may seem to be a quick solution to the out-of-reach problem. This, however, requires increased transmission power of the nodes leading to a shorter battery life of the network. It also increases the interference with neighbor nodes. It is well known from the Frii's path loss model of Equation (7) that the transmission power rapidly grows with the increasing distance. Equation (7) shows that the required $E_{tx}$ tends to grow excessively even for a small increase in distance. In this paper, therefore, we only consider a constrained distance and thus a limited transmission power for all nodes to preserve the battery lifetime.

### 3.2. Neighbor-Extended Maximal Connectivity Routing

To address the out-of-reach problem described above, we propose an enhanced recovery algorithm called Neighbor Extended MCR (NE-MCR), which is conducted after MCR. In addition to solving the out-of-reach problem of MCR, NE-MCR further reduces the energy consumption. It conducts an extended recovery process only for the nodes that failed to find their backup parent (BP) during their MCP process due to the out-of-reach problem. NE-MCR searches for a neighbor backup parent (NBP) in the neighborhood of the sibling nodes. Suppose that an $MCR\_BP$ has been chosen by MCR but has an out-of-reach problem. Then this $MCR\_BP$ broadcasts a special out-of-reach message to all its sibling nodes to trigger the NE-MCR algorithm. Then NE-MCR algorithm executes an individual search for the local optimum NBP in each sibling node of the $MCR\_BP$. Hence, $MCR\_BP$ node is an initiator of NE-MCR's procedure of selecting an NBP. In the end, $MCR\_BP$ receives the results of searches from the sibling nodes, compares them, and determines which NBP to choose as the optimum NBP. The detailed procedure of NE-MCR algorithm is presented below.

Procedure Select-NE-MCR_BP

For each parent node $p_j \in V$ of $G(V, E)$, if $MCR\_BP$ fails to reach grandparent $g$, repeat the following steps:

- For each child node $c_i$ of a parent node $p_j$:

    1. Select $c_k$ as a $MCR\_BP(p_j)$ if it has a maximal $Connectivity(c_k)$ with other siblings;
    2. $c_k$ informs its child nodes $c_j$ about the out-of-reach problem;
    3. Every $c_j$ examines each parent's neighbor $p_{ji}$ ($Distance(c_j, p_{ji}) < W_j$) for the following conditions:

        (a) Verify if the $Distance(c_j, p_{ji})$ is minimal;
        (b) Verify if the aggregated packet length $Length_\Sigma(l_{ji})$ of $p_{ji}$ is minimal;
        (c) Verify if $p_{ji}$ has an extra time slot $t_{ji}^{extra}$ to receive $c_j$'s data;

    4. If $p_{ji}$ best satisfies all three conditions, it sends $Optimal_{local}(p_{ji})$ to $c_k$;
    5. $c_k$ for each received $\{Optimal_{local}(p_{ji})\}$ from $c_j$:
    6. Determine $Optimal_{global}(p_{ji})$ with conditions (a), (b) and (c);
    7. If $Optimal_{global}(p_{ji})$ is determined by $c_j$ then assign $slot(p_j)$ to $c_j$;
    8. Else, keep $slot(p_j)$ assigning to $MCR\_BP(p_j)$;
    9. Send registration request message $M(c_j, slot(p_j))$ to $Optimal_{global}(p_{ji})$.

The operations of Procedure Select-NE-MCR_BP are described below for two cases:

- Case 1: parent node $p_j$ has only one child $c_1$.
- Case 2: $p_j$ has more than one child nodes $c_i$'s.

First, consider case 1. Let $p_j$ be the target parent node and let $c_1$ be a child node of $p_j$. Also let $g$ be the grandparent node of $c_1$. Suppose that $c_1$ cannot reach its grandparent $g$; in this case, NE-MCR carries out a single local search from $c_1$, and then it selects the optimum NBP within the wireless range of $c_1$. Then c1 sends a registration request message $M(c_1, slot(p_j))$ to the NBP. Here, $slot(p_j)$ indicates the time slot of $p_j$ allocated for its transmission in TDMA protocol. Then NBP registers $c_1$ with $slot(p_j)$, so upon the event when $p_j$ ever fails, NBP expects to receive data from $c_1$ not from $p_j$ during $slot(p_j)$. Like the MCR algorithm, for any pi that fails, the NE-MCR algorithm recycles $p_j$'s time slot for its child node $c_1$. Therefore, the fail recovery process requires no updates in time slot scheduling, leading to a low complexity and low power process. Additionally, it reduces the overhead of the whole fail recovery procedure. Otherwise, the child node should have sent a request for a new TDMA slot to NBP, and this also would have triggered the time-consuming process of rebuilding the time slot table for the entire network.

Now consider case 2, where parent node $p_j$ has more than one child nodes $c_i$'s. If the MCR algorithm finds no $MCR\_BP$ that can reach the grandparent $g$, it selects the $MCP\_BP$ with the maximum $Connectivity(c_i)$. In this case, the NE-MCR algorithm searches for neighbor backup parent NBPs in the neighbor subtrees.

For the child node $c_j$ (including $MCR\_BP$), NE-MCR conducts a search within the wireless range of $c_j$ and selects a NBP $p_{ji}$ that is a local optimum for $c_j$, if such an NBP exists. The objective function which evaluates the optimality of NBP is described in Section 4. Once $c_j$ determines its $Optimal_{local}(p_{ji})$, it forwards a message $M(c_j, p_{ji}, d_j)$ to $MCR\_BP$. The $MCR\_BP$ collects the messages on $Optimal_{local}(p_{ji})$ of all child nodes, and determines the globally optimum NBP ($Optimal_{global}(p_{ji})$). If the algorithm determines $Optimal_{global}(p_{ji})$ that is originally found by $c_j$, it assigns $slot(p_j)$ to $c_j$. Then, it requests $c_j$ to send $M_{ChildReq}$ message to $p_{ji}$. Like in Case 1, if $p_j$ ever fails in the forwarding operations for Case 2, $c_j$ uses $slot(p_j)$ to transmit data to $Optimal_{global}(p_{ji})$. Then, $c_j$ changes its role

from a child to $MCR\_BP$ and forwards the data from the isolated subtree under $p_j$ to $Optimal_{global}(p_{ji})$. If the $Optimal_{global}(p_{ji})$ is originally found by $MCR\_BP$, it keeps using $slot(p_j)$ and sends a registration request message to the $Optimal_{global}(p_{ji})$. Figure 6 illustrates a flow diagram of the overall NE-MCR algorithm, where the procedure Select-NE-$MCR\_BP$ is highlighted with a blue dotted line.

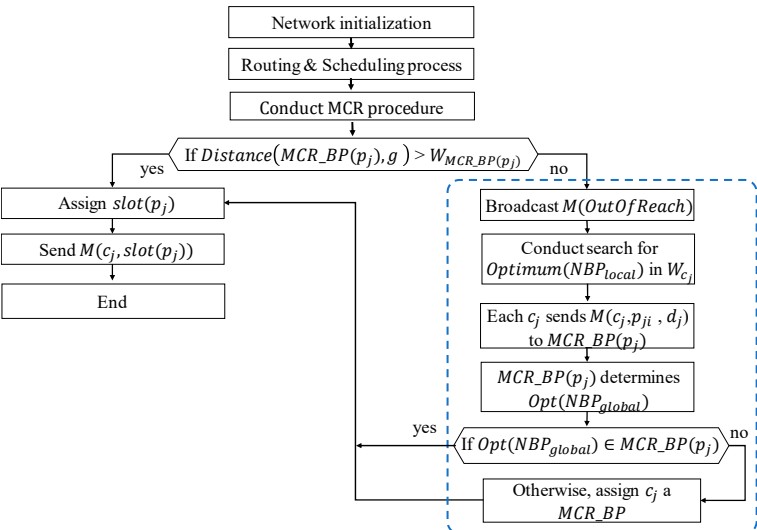

**Figure 6.** A flow diagram of proposed NE-MCR algorithm.

The NE-MCR algorithm conducts a search for $Optimal_{global}(p_{ji})$ only within the wireless range of $c_j$'s that are either $MCR\_BP$ or one of its siblings. The NE-MCR algorithm considers as candidates of NBP only the neighbor nodes that do not share the same parent with the current isolated nodes. For example, Figure 7 shows a subtree, where the target parent node is market by a red circle, while its child nodes are market by purple colors. Among the child nodes, a node is selected as $MCR\_BP$. For the target parent node, the MCR algorithm is conducted only by the purple nodes ($MCR\_BP$ and its siblings). The green nodes do not participate in the search operation, since they share the same parent (or grandparent) with the purple nodes. For candidates of NBP, only the blue nodes within the wireless range circles are eligible.

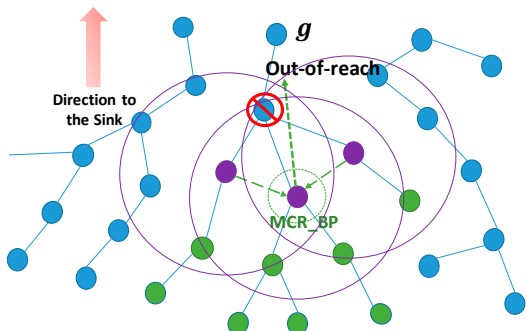

**Figure 7.** Avoid the same branch concept in NE-MCR algorithm.

To measure the optimality of candidate NBPs, we utilize the aggregated packet length of each NBP and the distance between $c_j$ and each NBP. In the current work, we assume that the selected NBP has an extra time slot available to receive additional data. When NE-MCR determines the $Optimal_{global}(p_{ji})$ with a closer distance to the child nodes $c_j$'s, it can reduce the transmission power of $c_j$'s and, therefore, can allow $c_j$'s to conserve more energy. Many literatures [24,35,38] have emphasized the critical effect of transmission distance on the energy consumption. To the best of our knowledge, however, no prior fault recovery methods like our method have been reported that minimize the distance from the

backup parent to the child nodes in the isolated subtree. In the following section, we discuss how we estimate the energy consumption during the search procedures for backup parents.

## 4. Constraints and Objectives of NE-MCR Algorithm

In this Section, we describe how the proposed network topology aggregates data and how the size of the aggregated data grows. Figure 8 illustrates an example subtree of a network that shows a data aggregation flow. In every active period, a node $n_i$ wakes up and obtains its sensing data $D_i$ of size S from its sensor. If $n_i$ is a leaf node, it forwards Ds to its parent node. If $n_i$ is a parent node, it receives a set of sensing data $D_k$ from all child nodes $n_k$. The parent node $n_i$ then aggregates the set of sensing data with its own sensing data. Finally, $n_i$ forwards the aggregated data to its parent node. For example, in the subtree, node n38 is chosen as a parent by two leaf nodes n41 and n42 nodes. We assume the sensing data generated by every node is of the same size S. Since n41 and n42 each send a datum of size S, n38 concatenates the two data and its own data into an aggregated packet of size 3S and forwards it to its grandparent n25. This Figure shows the size of data aggregated by every node in the above method.

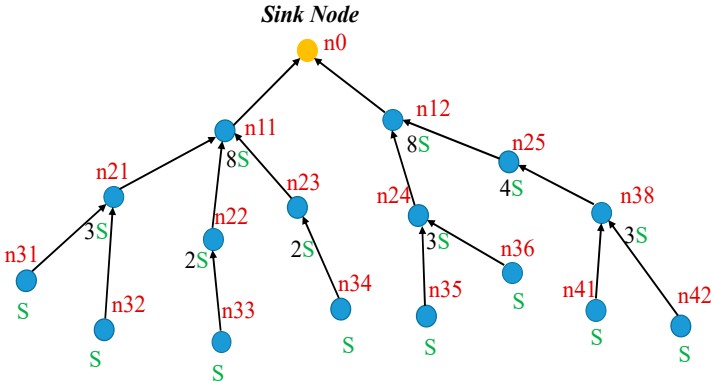

**Figure 8.** Data aggregation in tree-structured WSN.

As described in Section 2, in our network topology, every node is allocated with a TDMA time slot. Let $t_s$ be the fixed length of each time slot. This fixed slot length constrains the length of aggregated data in each node. For example, nodes n11 and n12 aggregate sensing data into a length of 8S each, which is the maximum data length in this subtree.

The length of each aggregated data must be shorter than the time slot constraint $t_s$. The routing algorithm selects route paths that meet this constraint. The proposed algorithm NE-MCR also ensures that this constraint is satisfied when it searches for the optimum NBP.

The constraint on the aggregated data length for NE-MCR is given by Equation (9).

$$l_{NBP_i} + l_{MCR\_BP} < t_s,\tag{9}$$

Here, $l_{NBP_i}$ is the length of $NBP_i$'s packet that is aggregated with the data received from all its child nodes. $l_{MCR\_BP}$ is the length of $MCR\_BP$'s aggregated packet that is forwarded from $MCR\_BP$ to $NBP_i$.

The energy model of tree-structured WSN comprises the sum of all transmission energy consumed by every child node $n_i$ and the sum of all reception energy consumed by every parent node while receiving data from its child nodes $n_i$. The energy model of a normal operation with no failed nodes can be formulated as follow:

$$E_{total} = \sum_{n_i \in N} E_{tx}(l_{n_i}, d(n_i, p_{n_i})) + \sum_{n_i \in N} E_{rx}(l_{n_i}),\tag{10}$$

Here, $N$ is the set of all nodes in the network, while $n_i$ represents every node in N. $p_{n_i}$ represents the parent node of $n_i$. $l_{n_i}$ denotes the length of data aggregated by node $n_i$. $d(n_i, p_{n_i})$ is the distance between $n_i$ and its parent $p_{n_i}$. $E_{tx}(l, d)$ is the transmission energy consumed by $n_i$ for transmitting a datum of length l through the distance d, while $E_{rx}(l)$ is the receiving energy consumed by a parent for receiving a datum from $n_i$.

In this work, for the sake of simplicity, we assume that the condition in Equation (9) is satisfied by the selected $NBP_j$'s parents $g_i^{NBP_j}$ and their successive parents who can receive additional data from the child nodes.

We now describe the energy model of a network with a fail recovery process for the case where node $n_i$ failed. Assume that the MCR algorithm selects $MCR\_BP_i$ as an MCR backup parent node for $n_i$. Assuming that $MCR\_BP_i$ cannot reach the parent of pi, now suppose that the NE-MCR algorithm selects $NBP_i$ as a neighbor backup parent. When failure occurs at $n_i$, the new data forwarding recovered by using the preselected $MCR\_BP_i$ and $NBP_i$ incurs variable transmission energy $E_{TX_i}^{new}$ which can be expressed by Equation (11).

$$E_{TX_i}^{new} = E_{tx}^{BP_i \rightarrow NBP_i} + \sum E_{tx}^{C_m \rightarrow BP_i}, \tag{11}$$

Here, $E_{tx}^{BP_i \rightarrow NBP_i}$ indicates the new transmit energy of a recovery route from $MCR\_BP_i$ to $NBP_i$. The second term $\sum E_{tx}^{C_m \rightarrow BP_i}$ accounts for the total transmit energy of all other child nodes Cm forwards their data to $MCR\_BP_i$ nodes. Using Equation (7), we can rewrite Equation (11) by Equation (12).

$$E_{TX_i}^{new} = l_{MCR_{BP_i}} E_{elec} + (l_{c_1} E_{elec} + l_{c_2} E_{elec} + l_{c_3} E_{elec} + \ldots + l_{c_m} E_{elec}) + $$
$$l_{MCR_{BP}} E_{FS} d_{MCR_{BP_i}, NBP_i}^2 + (l_{c_1} E_{FS} d_{c_1}^2 + l_{c_2} E_{FS} d_{c_2}^2 + l_{c_3} E_{FS} d_{c_3}^2 + \ldots + l_{c_m} E_{FS} d_{c_m}^2), \tag{12}$$

In Equation (12), $E_{elec}$ is the unit energy per data bit consumed by the transceiver circuit. This paper assumes that $E_{elec}$ is constant for all nodes. $l_{MCR_{BP_i}}$ denotes the length of the packet that $MCR\_BP_i$ forwards to $NBP_i$, whereas $l_{C_m}$ indicates the length of the packet that the other child nodes $C_m$ forwards to $MCR\_BP_i$. $d_{MCR_{BP_i}, NBP_i}$ denotes the transmission distance from $MCR\_BP_i$ to $NBP_i$, while $d_{c_m}$ indicates the distance from $C_m$ to $MCR\_BP_i$. Since the first two terms are constant, it can be substituted by $C_{elec}$, so Equation (12) is simplified by Equation (13).

$$E_{TX_i}^{new} = C_{elec} + E_{FS}(l_{MCR_{BP}} d_{MCR_{BP_i}, NBP_i}^2 + \sum_{1 \leq m \leq M} l_{C_m} d_{C_m}^2), \tag{13}$$

Here, $E_{FS} l_{MCR_{BP}} d_{MCR_{BP_i}, NBP_i}^2$ denotes the transmission energy of the link from $MCR_{BP_i}$ to $NBP_i$, whereas $E_{FS} \sum_{1 \leq m \leq M} l_{C_m} d_{C_m}^2$ represents the sum of transmission energy from all Cm's to $MCR_{BP_i}$.

Using Equation (13) as the cost function, the objective formula of the proposed algorithm NE-MCR is given by Equation (14) under the constraints given by Equations (15)~(17). For all nodes $n_i \in N$, it selects a set of backup pairs $(MCR\_BP_i, NBP_i)$ that minimize the cost function $E_{TX_i}^{new}$, respectively for each $n_i$.

Objective:

Minimize $E_{TX_i}^{new}$ while selecting $MCR\_BP_i$ and $NBP_i$ for every node $n_i \in N$ (14)

Such that:

$$l_{NBP_i} + l_{MCR\_BP_i} < t_s, \tag{15}$$

$$d(C_m, MCR_{BP_i}) \leq W, \tag{16}$$

$$d(MCR_{BP_i}, NBP_i) \leq W, \tag{17}$$

Equation (15) defines the constraint that the aggregated data length of $NBP_i$ must not exceed a threshold $t_s$ as presented in Equation (9). Equation (16) stipulates that the child node considered as $MCR\_BP_i$ must be reachable from all other child nodes $C_m$ with the wireless range $W$, whereas

Equation (17) stipulates that $NBP_i$ must be reachable from the selected $MCR\_BP_i$. In this way, the NE-MCR algorithm finds an optimal backup pair ($MCR\_BP_i$, $NBP_i$) that meets the optimization objectives and constraints given by Equations (14) and (17).

For example, Figure 9 illustrates a subtree of a network to depict how NE-MCR selects an optimal pair of $MCR\_BP_i$ and $NBP_i$ for a node $n_i$. NE-MCR repeats this selection process for every node $n_i \in N$ ($N$-the total number of node in a network) as a process to find a recovery route path for the case where $n_i$ indeed fails during normal operation. In Figure 9, the potential faulty node $n_i$ is highlighted. In this subtree, none of its child node $C_m$'s can reach their grandparent $g$, and thus the MCR algorithm fails in finding a backup parent. Therefore, NE-MCR attempts to find an optimal pair ($MCR\_BP_i$, $NBP_i$) as follows. NE-MCR checks the potential of each $C_m$ and adds $C_m$ to the candidate set of $MCR\_BP_i$, if $C_m$ meets the constraints of Equation (16). For each $MCR\_BP_i$ of the candidate set, NE-MCR finds a set of $NBP_i$ nodes that satisfy the constraints of Equation (17), and adds a pair ($MCR\_BP_i$, $NBP_i$) to a set of candidate pairs. NE-MCR then calculates the cost function $E_{TX_i}^{new}$ of every candidate pair, and selects the pair ($MCR\_BP_i$, $NBP_i$) of the minimum $E_{TX_i}^{new}$ as the optimal recovery backup nodes.

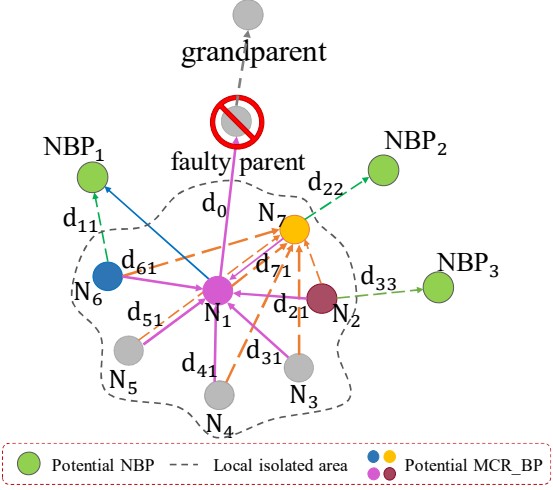

**Figure 9.** Example sub-tree of network where NE-MCR selects an optimal NBP.

## 5. Results and Discussion

In this section, we discuss the performance analysis of our fail recovery routing approach. To evaluate, we compare the simulation results of the proposed method with the existing ESCFR and DCFR algorithms. These algorithms forward the data of sensor nodes to the base station using a backbone formed by a particular set of nodes. The nodes in the backbone are selected based on a cost function. Before the actual data forwarding, each node uses the cost function to identify the minimum power for the current transmission and the neighbor node with the maximum remaining energy. This backbone can be changed at any time if the result of the cost function becomes less optimal for the corresponding chain of the backbone.

Many previous articles such as References [21–24] report that the network lifetime drastically changes when the variation in the number of nodes or in the node's transmission range occurs. Therefore, during the simulation, we use these network parameters as a varying argument.

### 5.1. Analysis of the Performance of NE-MCR

In order to generate simulation results, we exploited the C++ program based on the WiSer simulation tool introduced by [27]. This tool first generates a spanning tree of the target network by conducting a routing algorithm that is presented in [27]. For example, Figure 10 illustrates such a spanning tree. Then the simulator allocates TDMA time slots to each node in the spanning tree using a multichannel and multi-hop scheduling algorithm presented by [5].

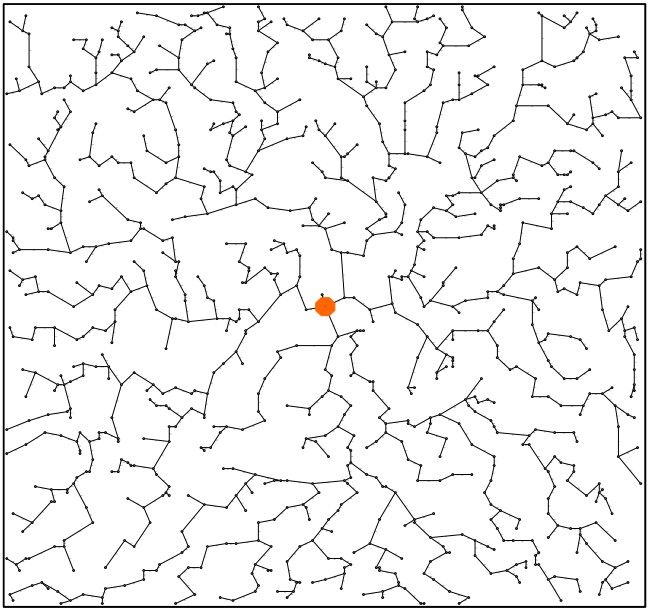

**Figure 10.** Spanning tree-based routing-enabled WSN generated in WiSer C++.

To examine the reliability of the proposed algorithm, we conducted simulations using example networks of different density varying from 100 up to 1000 nodes for a 1000 m $\times$ 1000 m area. The sink node was placed in the center of the given area. We injected faults to 10 percent of the nodes to evaluate the network connectivity ratio of the MCR, DCFR and NE-MCR algorithms, respectively. The network connectivity ratio $\beta_c$ is defined as follow:

$$\beta_c = M/N \tag{18}$$

Here, $M$ is the number of nodes that can forward the data to the sink node and $N$ denotes the total number of nodes in the network.

Figure 11a shows the network connectivity ratio of the proposed algorithms, and compares algorithms for 10 example networks with 100 nodes up to 1000 nodes. The NE-MCR algorithm provides an increasingly higher network connectivity ratio for denser networks. Similarly, DCFR algorithm also performs a better network connectivity ratio, but we can see fluctuating behavior as well for the denser networks such as from 400 to 800. On the other hand, MCR algorithm shows the network connectivity ratio decreasing as the network density grows beyond 300 nodes.

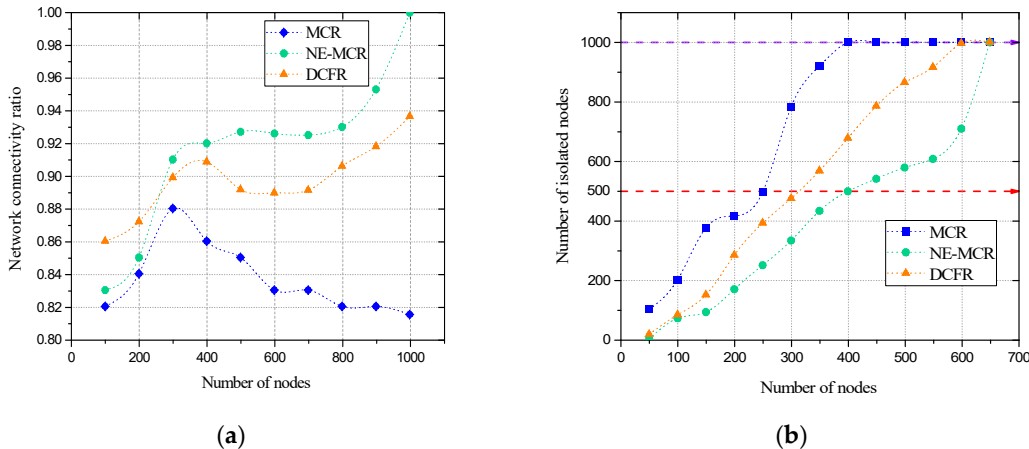

**Figure 11.** Simulation results regarding: (**a**) network connectivity ratio; (**b**) network isolation rate.

Another set of experiments are conducted to evaluate the recovery capability for an increasing number of faults. For all algorithms we increased the number of faulty nodes until the point where the network reaches a complete halt. Figure 11b shows the number of isolated nodes with a lost connection when the number of fault injections increases for the network of 1000 nodes. We compare two network conditions: A half-network-isolated condition (red dashed line) and a whole-network isolated condition (purple dashed line).

Figure 11b shows that NE-MCR reaches the half-network-isolated condition when the fault injection ratio is 40% (400 faulty nodes out of 1000), whereas DCFR and MCR reach this condition much earlier when the fault injection ratio becomes 32% and 24% respectively. For NE-MCR, the whole-network-isolated condition comes as late as the fault injection ratio of 65%. In contrast, for MCR, this condition is reached as early as the fault injection ratio of 40%. DCFR algorithm results in the whole-network-isolated condition when the number of injected faulty nodes accounts for 60% of the overall network nodes. This experiment demonstrates that NE-MCR sustains the operation for the rest of the network significantly longer than DCFR and MCR algorithms. The worst result is produced by the MCR which proves that this algorithm alone cannot solve the out-of-reach problem. NE-MCR and DCFR algorithms can find alternative or backup recovery solutions for most of the faulty nodes that have the out-of-reach problem.

### 5.2. Energy Efficiency of NE-MCR

To evaluate the energy efficiency of the proposed algorithm, we compare the network lifetime and energy consumption of NE-MCR with DCFR and ESCFR methods. In this experiment, we assume that the only cause of node failure is a dead battery for the sake of focus on the energy efficiency. For each example network, the simulation initiates the network operation by filling every node with full battery energy. As each node starts forwarding the aggregated data towards the base station, it gradually drains its battery using the energy model of Equation (6). Table 1 summarizes the unit energy parameters used by the energy model [11]. The proposed method is a route recovery method, not a node recovery. Thus, the FDT of the network is not relevant to our performance evaluation.

**Table 1.** Default value of all network parameters used in the simulation.

| Parameter | Value |
|---|---|
| $E_{elec}$ | 50 (nJ/bit) |
| $E_{Fs}$ | 10 (pJ/bit/m$^2$) |
| $E_{amp}$ | 0.0013 (pJ/bit/m$^4$) |
| Initial energy | 0.5 (J) |

On the other hand, the ADT is a suitable performance metric. Figure 12a compares the ADT performance of all three fault recovery routing methods, NE-MCR, DCFR and ESCFR, for the 10 network examples of Figure 11a. NE-MCR and ESCFR methods show that the ADT decreases when the number of nodes increases. Surprisingly, DCFR experiences an increase when the number of nodes grows from 400 to 500. This algorithm uses a different cost function than ESCFR and it periodically updates information regarding the available energy of all neighbor nodes. Therefore, it balances a network load more effectively than ESCFR by changing the set of nodes in backbones. In terms of NE-MCR, as the number of child nodes increases, the backup parent nodes receive more data from their child nodes and consume higher energy. Nevertheless, it still performs better ADT, since it also balances the network load considering transmission power and NBP's receiving energy during the recovery procedure.

Due to the random injection of a fault, some nodes may not find closer neighbor nodes to choose as an NBP, and thus this drains their battery faster. Those nodes, therefore, may spend more transmit energy since the distance between $MCR\_BP_i$ and optimal $NBP_i$ is greater than others.

Figure 12a shows that until the number of network nodes reaches 400 (red dashed line), the ADT of NE-MCR is substantially greater than the ADT of the other two algorithms. Because, until this point, the network density is low and the nodes using DCFR or ESCFR are more likely to choose a father node that has more available energy to forward their data. As the number of nodes increases, a difference of ADT between these methods shrinks. However, we can observe that for a lower density network cases, NE-MCR algorithm improves ADT by 21% on average over the compared methods. Figure 12b compares the average energy consumption per node for all fail recovery routing methods. The average energy consumption per node grows gradually as the network density increases, and so more data is aggregated in each node. For the network of 200 nodes, our proposed NE-MCR algorithm consumes around 40% less energy than the other two reference algorithms. However, for higher density networks, NE-MCR algorithm's energy consumption grows rapidly and a difference in performance of the proposed method with DCFR shrinks up to 16% (for the network with 500 nodes). We can see some non-linearity in the performance of DCFR whereas ESCFR experiences a linear increase as the network density grows.

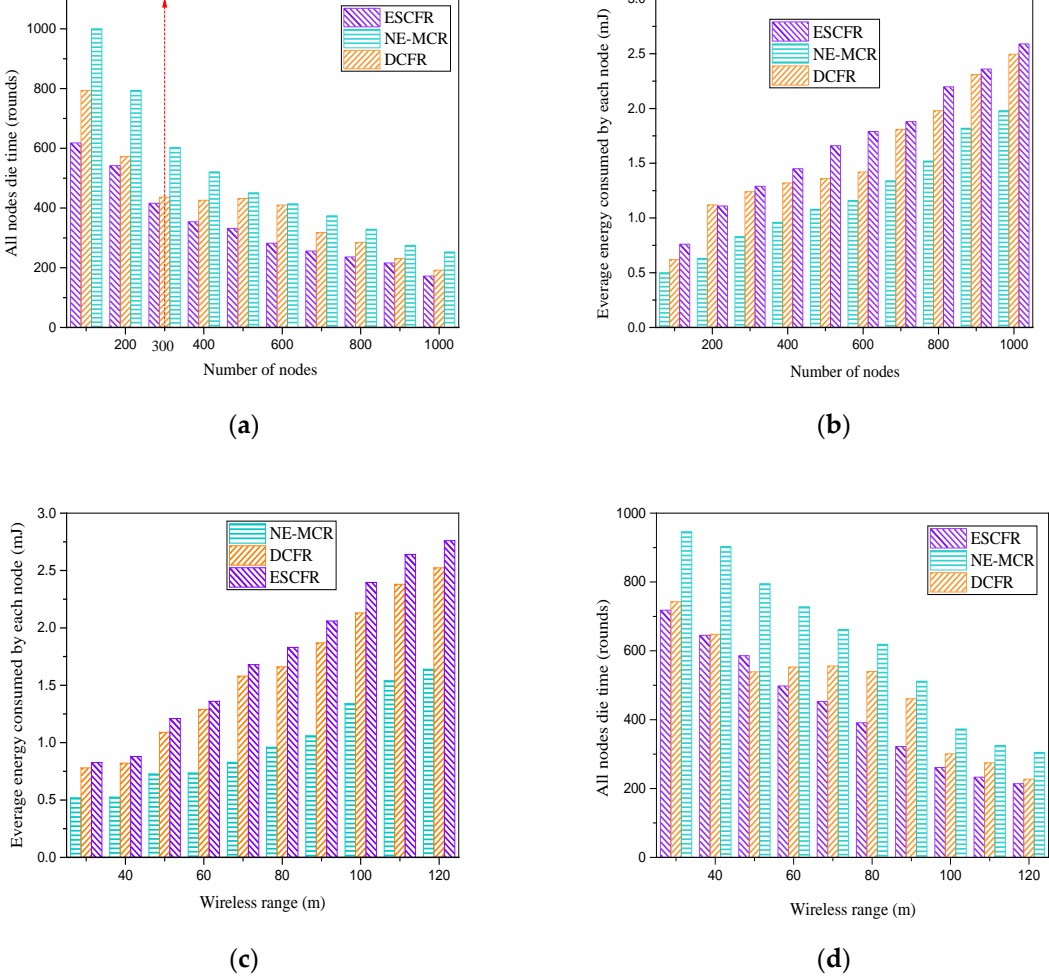

**Figure 12.** Simulation results for energy efficiency considering: (**a**) ADT versus network density; (**b**) energy consumption versus network density; (**c**) energy consumption versus wireless range; (**d**) ADT versus wireless range.

Figure 12c compares the average energy consumption per node measured with various wireless ranges for each node. While all three algorithms have increasing energy consumption as the wireless range grows, NE-MCR shows substantially lower energy consumption in all wireless ranges tested.

For example, for the wireless range of 90 m, NE-MCR consumes about 40% lower energy than DFCR. For the same wireless range, ESCFR consumes 48% more energy than the proposed method.

Figure 12d illustrates ADT measured over various wireless ranges for the proposed NE-MCR and comparing DCFR and ESCFR algorithms, respectively, for an example network of 600 nodes. The nodes that transmit data to a longer distance consume higher energy and, therefore, drain their battery faster. NE-MCR shows better performance due to its key advantage that minimize the transmit distance for backup parents, while balancing the number of child nodes to meet the data size constraints. Consequently, Figure 12d demonstrates that NE-MCR performs around 30% more data collection rounds than the compared algorithms for the network with a 40 m wireless range.

## 6. Conclusions

This paper presented an energy-efficient fail recovery routing algorithm targeted for a tree-topology wireless network whose nodes can fail due to battery depletion. When the recovery process is initiated, MCR algorithm determines a back-up node for each parent from a local subtree. Then, this back-up node employs faulty parent's TDMA slot to forward aggregated data of the subtree to grandparent. In the implementation stage, we observed that some back-up nodes are not able to connect their grandparents due to a distance constraint. We increased the transmission power of back-up nodes and then they were capable of forwarding their data to the grandparents. This small modification of transmission power, however, caused the following problems: (a) The nodes who were out of the interference zone of the back-up node started facing collision if their slots were identical, and (b) the back-up nodes were identified as faulty nodes in further steps since they used higher power to execute each transmission. Thus, we applied our second NE-MCR recovery method to find back-up parent nodes from different branches of the spanning tree. In this phase, we faced other constraints: (c) The back-up parents selected by NE-MCR were only able to accept a limited number of child nodes; (d) slot length was constrained and it was assigned in an earlier scheduling phase. Allocation of longer slots causes additional energy consumption due to the Idle mode of parents who have less child nodes. However, in a denser network scenario, more back-up patents were found, and NE-MCR was able to connect the isolated subtree with the optimal back-up parent. We compared the proposed method with reference algorithms in a wide range of network sizes. In comparison with reference algorithms, NE-MCR provided a substantially higher network connectivity ratio for networks of greater than 400 nodes. When compared with ESCFR and DCFR algorithms, NE-MCR consumed on average 23% less energy, while allowing a 21% longer lifetime for large networks. The proposed algorithm, therefore, is well suited to a fast recovery solution for low power networks. In addition, it offers a non-disruptive recovery solution for TDMA networks since it finds all back up parents without changing the existing scheduling.

**Author Contributions:** Conceptualization, O.U. and H.W.K.; Methodology, H.W.K.; Software, O.U.; Validation, O.U., H.W.K.; Formal analysis, H.W.K.; Writing—original draft preparation, O.U., H.W.K.; writing—review and editing, H.W.K.; funding acquisition, H.W.K.

**Acknowledgments:** This work was financially supported by the Center for Integrated Smart Sensors funded by the Ministry of Science, ICT & Future Planning as Global Frontier Project, Korea (CISS-2017M3A6A6066117). This work was also supported by IITP grant funded by the Korean government (No. R7117-19-0164, Development of wide area driving environment awareness and cooperative driving technology which are based on V2X wireless communication). It was also supported by SoC platform and SW development Advanced Education for IOT which is funded by the Ministry of Trade, Industry, and Energy (N00011132).

**Conflicts of Interest:** The authors declare no conflict of interest.

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
