# Peer review of "An Energy-Efficient Fail Recovery Routing in TDMA MAC Protocol-Based Wireless Sensor Network"

_electronics, doi:10.3390/electronics7120444_

Round 1

Reviewer 1 Report

The remarks can be found in the file.

Author Response

Reviewer: 1

The paper proposes an energy-efficient fault recovery in a wireless sensors network. While the paper is potentially interesting, there are several aspects that should be clarified by authors.

[Reviewer 1, Comment #1]

Why do we need WSNs nowadays?

[Author’s Response to Comment #1]

      Apologies for giving less background information regarding the importance of WSNs. We added following paragraph   

[The following sentences has been added/modified to the manuscript]  

On Page 1 Line 35.

     “Sensor nodes are used widely in industry to monitor and accumulate the data related to the object. For instance, deploying sensor nodes, we can receive periodic information about the environments such as wild nature (forests or deserts), special industrial zones etc. [2]. We may apply WSN to obtain an up-to-date temperature information or monitor toxic gas level in different branches of industry. Large-scale self-organized wireless sensor and mesh network provides an opportunity to develop Smart Environment and Smart Grids applications [1].  The WSN is critically important to support these advanced applications.”

[Reviewer 1, Comment #2]

Why the battery depletion cannot be solved by adding larger batteries? Maybe the characteristic of the WSNs, in particular of sensors and their constraints may be more emphasized.

[Author’s Response to Comment #2]

    Thanks for your kind suggestion. We modified the manuscript to incorporate your suggestion.  

[The following sentences has been added/modified to the manuscript]  

On Page 4 Line 161

    “In most environmental monitoring, infrastructure or facility diagnosis wireless metering applications, employing energy harvesting devices like solar cell is not an effective solution since sensor devices are usually installed in indoor environment or dark basements of the buildings. Using a larger battery is not acceptable due to the cost and size constraints on the sensor devices, since these applications are often deployed throughout the entire city to monitor temperature, air pollution or toxic gas level.”

[Reviewer 1, Comment #3]

line 159: you mention that each node wakes up at the same time. May this cause lack of samples for the reconstruction of the sensed event?

[Author’s Response to Comment #3]

Nowadays the sensor nodes in most IoT applications use special camera to capture the image of object. It extracts the data using the captured image. For instance, in Smart Grids and Wireless Metering applications image recognition algorithms are employed to extract the data correctly using the captured image of sensor’s camera.

[Reviewer 1, Comment #4]

Fig. 2 may be briefly explained.

[Author’s Response to Comment #4]

     Thank you for your suggestion. We added brief explanation of Figure.2 in the main manuscript.

[The following sentences has been added/modified to the manuscript]  

On Page 5 Line 194

     ” In this network scenario, a sensor node first selects its parent node, and then schedules its transmission in the time slot that is earlier than the slot of selected parent. Employing different RF channels allows concurrent time slots for large-scale networks. It also mitigates the interference between the nodes that are using identical slots in the zone of interference. In Figure.2, node n31 and n33 select the same time slot. Since they use different channels, their concurrent transmission does not cause collision. Although each sensor node has a single radio to communicate, it can bridge the child nodes that use various RF channels with the sink node.  Initially, it tunes the RF channel of child node who is allotted with the earliest slot. Then, it switches to other channels according to the sequence of the slots assigned to child nodes. The time consumed for switching from one channel to another is negligible [18].  For instance, in Fig. 2, node n11 receives data from child nodes using channel 1 and 2. Then, it forwards the aggregated data to the sink node. The process of data forwarding in various time slots is depicted in Fig. 2(b)”.

[Reviewer 1, Comment #5]

In Fig. 4, letters are used to identify each node, whereas in Fig. 2,3 numbers are used. Maybe

always the same notation may be used.

[Author’s Response to Comment #5]

      In Figures 2 and 3, we gave brief examples of small network segments to make it easier for the readers to understand the goals. In Fig. 4 we tried to illustrate the selection of MCR_BP in common network. Some notation of Fig.4(a) is used in the flow diagram of Fig.4(b). In the main procedures of algorithm, it is suggested to use common notations that represents abstract nodes. As you said, using similar notations make the examples more conceivable to readers.  This is the reason we used the same notations in the examples in Figure 2 and 3. 

[Reviewer 1, Comment #6]

the fault recovery is explained. A node  is selected as MCR_BP. It will be the parent of its brothers when its parent exploits its battery. What about s brothers who cannot

reach ? You mention later (line 479-480 and Eq. 16) that MCR_BP has to be reachable from all its brothers. Maybe this should be specified also before.

[Author’s Response to Comment #6]

      Apologies for making you confused. In the first recovery process, we try to select the one that has maximal connectivity with its own siblings as an MCR_BP. As you point out,   may not cover all the siblings. In the simulation stage, we experienced some nodes isolated due to large distance with elected MCR_BP. However, the number of such nodes decreased as the density of the network increases.  We added more details regarding this special case in Section 3.

[The following sentences has been added/modified to the manuscript]  

On Page 10 Line 3657

The proposed recovery algorithm selects the node that has maximal connectivity with other siblings. In sparse network scenario, however, the elected  may not cover all siblings. Therefore, in Figure 5, our method experiences additional isolated nodes when the number of induced faulty node grows.

[Reviewer 1, Comment #7]

Some additional information about fault detection may be given (only mentioned in the introduction).

[Author’s Response to Comment #7]

Following paragraph is added to give additional details regarding failure detection.

[The following sentences has been added/modified to the manuscript]

On Page 4 Line 135

“In WSNs, a number of different techniques can be applied to detect the failures. Some researchers proposed the use of passive information collection for the purpose of failure detection. In these methods, information crucial to detect the failure can be extracted from regular data packets sent to the sink node [31-32]. In [30], the authors proposed a special framework to detect the failure in WSN. In this model, sensor nodes piggyback path checksum tags upon all regular messages sent to the sink node. Each node updates the tags with its own node ID by means of Fletcher checksum algorithm. After receiving packet from all routes, the sink node inspects their checksum. To identify any changes in specific path, the sink node injects series of control messages. Based on the response to these messages, the sink determines and reports the failure. Most of failure detection algorithms use additional control information which incur overhead in low power WSN. Therefore, in current work, faulty parents are detected by identifying the nodes that do not acknowledge their child within a given period”.

[Reviewer 1, Comment #8]

I think that a full stop should be added in line 310.

[Author’s Response to Comment #8]

Thanks for noticing this error, we added the full stop to corresponding sentence (which is now line 359).

[Reviewer 1, Comment #9]

in line 319, you mention that the BP selection occurs only when the network is initialized. When a node  is selected as MCR_BP and it becomes the parent of its brothers because of its parent fault, why does a new MCR_BP selection not occur?

[Author’s Response to Comment #9]

We targeted the application that allows a sensor node to stay awake only in very short time slots. A new MCR_BP selection process requires additional time to exchange control messages of recovery algorithm. If they stay longer and transmit recovery control messages, they may violate the scheduling table and cause interference to other nodes that use later time-slots. Launching re-scheduling process due to some faulty nodes triggers significant overhead and it causes increase energy consumption. Therefore, we considered proactive recovery procedure during the network initialization phase.

[Reviewer 1, Comment #10]

line 349: what do you mean withneighbor parent?

[Author’s Response to Comment #10]

Thanks for pointing it out. After your remark, we changed “neighbor parent” to parent’s neighbor which is a parent node of different branch. Sometimes within the wireless range of child node, there can be multiple parent nodes that are neighbor with primary parent.

[Reviewer 1, Comment #11]

line 358: who is p?

[Author’s Response to Comment #11]

Thanks for your remark. It should   that represents selected backup parent node using MCR algorithm. We gave brief explanation in line 402, 411. We corrected this spelling error in main manuscript.

[Reviewer 1, Comment #12]

line 369: does NBP expect to receive data from  during ? The sentence is not clear. Moreover, does NBP always (for all the ADT) expect to receive data from c1 during ? Does this cause a waste of energy since unused time slots are needed?

[Author’s Response to Comment #12]

As I mentioned, all nodes transmit the data according to the strict scheduling table. Therefore,   has to switch its TX slot to the faulty parent  ’s time slot in order to abstain NBP from rescheduling procedure. Suppose, if   keeps using its own slot, then it should make sure that during this slot NBP is not receiving data from any other child node. To make it safe,   needs to change its slot to  ’s slot.

[Reviewer 1, Comment #13]

in Fig. 6, the yes/no tags in the second if condition are missing.

[Author’s Response to Comment #13]

We appreciate your remark. we added yes/no tags in Figure 6.

[Reviewer 1, Comment #14]

line 379-391: I am not sure to have correctly understood the described procedure. Reading the text, it seems that if  proposes the best , it becomes the MCR_BP. The parent of  and of its brothers becomes . How do you manage the situation in which the  brothers cannot reach ? Maybe you mean that if  proposes the best , it becomes the parent of its brothers. The parent of is now .          

[Author’s Response to Comment #14]

Apologies for the confusion. As you mentioned in the previous comment,   proposes the best  , which becomes the parent of its brothers. Then,   becomes the parent of  .   sends child request message ( ) to register in the local scheduling table of   .

[Reviewer 1, Comment #15]

When is the NE-MCR procedure performed (e.g. in the initialization network phase)?

[Author’s Response to Comment #15]

We perform NE-MCR in the network initialization phase, when routing and scheduling are completed. NE-MCR triggers whenever MCR_BP encounters out-of-reach problem that is defined in line 346 of main manuscript. We have mentioned thin in Figure 6.

[Reviewer 1, Comment #16]

line 417: I think that withgrandparentyou mean theparentof . Otherwise, a node forwards data directly to its grandparent.

[Author’s Response to Comment #16]

Actually, using “grandparent” may confuse the readers. Thanks for your remark, we changed it to “parent” of    .

[Reviewer 1, Comment #17]

line 486: what is ?

[Author’s Response to Comment #17]

Here,   is the total number of nodes in the network. We added brief details.

[The following sentences has been added/modified to the manuscript]  

On Page 16 Line 539

( –the total number of nodes in the network)”.

[Reviewer 1, Comment #18]

A definition of network connectivity ratio may be added.

[Author’s Response to Comment #18]

We added a brief description of network connectivity ratio in Section 5.

[The following sentences has been added/modified to the manuscript]

On Page 17 Line 571

“The network connectivity ratio   is defined as follow:

                                    (18)

here,   is the number of nodes that can forward the data to the sink node and   denotes the total number of nodes in the network”.

General remarks

 [Reviewer 1, Comment #19]

You give a definition of MCR_BP in section 3: in case of parent failure, the node, which was elected as MCR_BP, becomes the parent of its brothers. In the paper, I think that sometimes you use MCR_BP instead of parent and vice versa.

[Author’s Response to Comment #19]

Thanks for your remark. After Section 3, we mostly analyzed the event when MCR_BP is encountered with out-of-reach problem. The NE-MCR algorithm is specially designed to solve this problem. While NE-MCR was executed, primary parent had already become faulty and MCR_BP had been also selected to substitute the faulty parent. Therefore, we often referred to MCR_BP instead of its faulty parent.      

[Author’s Response to Comment #20]

Be careful to the use of acronym and the full names (for example line 151-152: WSN is written in full).

[Author’s Response to Comment #20]

Thanks for your kind remark. We changed all full names such as wireless sensor network with initially defined WSN. In the abstract line 14, in introduction line 30 and 68, in the caption of Figure 1 and Figure 2 and Section 2, line 159-160.

[Author’s Response to Comment #21]

If more coherence is used for the notation, the reader can easier understand your work: indicates child nodes, parent nodes, , etc., always in the same way (e.g. the same letter,  or  or ).

[Author’s Response to Comment #21]

As you said, the coherence is very crucial to make the text and the process details more straightforward to the readers. We agree with your suggestion. However, in some paragraphs of Section 2, we applied some notations from a graph theory to make it easier for the readers to understand the procedures. Only in these paragraphs, we used   to indicate the backup parent that is determined by MCR algorithm. As you indicated correctly, in Section 4, we applied different notation for the same terms. As a result, we changed all   to  .

Reviewer 2 Report

The paper is interesting and well presented. Results show the benefits of the proposal.

Some minor corrections or clarifications could be done:

In section 2 authors mention the use of a set of RF channels in the scheduling process, but following sections do not mention noting about it. A clarification must be done of the implications of using different RF channels (if possible) in your proposal.

Subsection 2.2 explain that authors assume that every node is assigned a constrained transmission power. From an implementation point of view (not in simulation) an in-depth explanation of this item must be done.

Section 3:

The propose route algorithms assume a bidirectional communication between child nodes and parent. It must be clarified how I can be done in a TDMA scheme. How information can be broadcasted in a TDMA (from an implementation point of view) with probably different channels? It is a reliable broadcast?

Additionally it must be enumerated the requirements of using the proposal: it can be used in a unidirectional WSN where data only flows from sensor nodes to sink nodes (or parents)? 

How a failed node can be detected by the others? As different WSN technologies and protocols exsist, which of them are valid for using your proposal? 

If a failed node is restored (by example because the use of energy harvesting techniques charge the battery some time later), how does the protocols respond?

About isolated nodes authors compares with a naïve recovery method (based on random selection) Why? Other alternatives?

Finally an study of the data length and the time slot should be interesting. How it is defined the time-slot?  The relation between the time slot duration and the number of childs and data length should be studied. What happens if childs data length exceeds the slot duration and no more parents are available? How this situation is treated by the algorithms?

Author Response

Reviewer: 2

The paper is interesting and well presented. Results show the benefits of the proposal. Some minor corrections or clarifications could be done:

[Reviewer 2, Comment #1]

In section 2 authors mention the use of a set of RF channels in the scheduling process but following sections do not mention noting about it. A clarification must be done of the implications of using different RF channels (if possible) in your proposal.

[Author’s Response to Comment #1]

We appreciate your comment and apologies for lack of information regarding the exploitation of various RF channels. The network model of current work is explained in detail in the reference [18]. Furthermore, we added more details about the benefits of using different RF channels in the manuscript.

[The following sentences has been added/modified to the manuscript]

On Page 5 Line 193

In this network scenario, a sensor node first selects its parent node, and then schedules its transmission in the time slot that is earlier than the slot of selected parent. Employing different RF channels allows concurrent time slots for large-scale networks. It also mitigates the interference between the nodes that are using identical slots in the zone of interference. In Figure.2, node n31 and n33 select the same time slot. Since they use different channels, their concurrent transmission does not cause collision. Although each sensor node has a single radio to communicate, it can bridge the child nodes that use various RF channels with the sink node.  Initially, it tunes the RF channel of child node who is allotted with the earliest slot. Then, it switches to other channels according to the sequence of the slots assigned to child nodes. The time consumed for switching from one channel to another is negligible [18].  For instance, in Fig. 2, node n11 receives data from child nodes using channel 1 and 2. Then, it forwards the aggregated data to the sink node. The process of data forwarding in various time slots is depicted in Fig. 2(b)”.

[Reviewer 2, Comment #2]

Subsection 2.2 explain that authors assume that every node is assigned a constrained transmission power. From an implementation point of view (not in simulation) an in-depth explanation of this item must be done.

[Author’s Response to Comment #2]

We appreciate your remark. As we indicated in Eq. (7), transmission distance and data length are main argument of transmission energy function. These two parameters are proportional to the energy consumption. Suppose that initially we set transmission power of node-A approximately 10 dBm and we aim to cover 150 m range with 95 % PDR. Node-A may create 200 m round interference zone for other neighbor nodes. It means, the other nodes within 200 m round should not execute transmission while node-A sends data to its parent. However, nodes within 250 m may execute transmission since they are out of interference zone of node-A. If we increase the TX power of node-A to 15 dBm, its interference zone obviously enlarges, and it may interfere the nodes within 250 m range. On the other hand, now node-A consumes 5 dBm more power for each transmission. Then, the battery with limited energy of node-A suffers from intense drainage which degrades lifetime of node-A. Therefore, in strictly scheduled TDMA network, we cannot simply increase the TX power which is constrained by above reasons.       

[The following sentences has been added/modified to the manuscript]

On Page 7 Line 255

“The transmission distance and packet length are main arguments of transmission energy function. These two parameters are proportional to the energy consumption. Let’s suppose that initially, node-A’s transmission power is set approximately 5 dBm and it is aimed to cover 150 m range with 95 % PDR. Node-A may create 200 m round interference zone for other neighbor nodes that use the same slot. Hence, these nodes should not execute transmission when node-A sends its data to the parent. However, nodes within 250 m are allowed to transmit data using the same slot since their transmission is not interfered by node-A. If node-A increases its transmission power to 10 dBm, then, its interference zone obviously enlarges, and it interferes the nodes within 250 m range. On the other hand, now node-A consumes twice more energy for each transmission and its battery suffers from intense drainage of energy. Thus, in strictly scheduled TDMA MAC protocol based WSNs, we cannot merely increase power of transmission due to the above constraints”.

[Reviewer 2, Comment #3]

The propose route algorithms assume a bidirectional communication between child nodes and parent. It must be clarified how I can be done in a TDMA scheme. How information can be broadcasted in a TDMA (from an implementation point of view) with probably different channels? It is a reliable broadcast?

[Author’s Response to Comment #3]

In TDMA MAC protocol based WSNs, every sensor node is allotted unique time to transmit its sensed data to selected parent node. As we mentioned in the lines 164-166.

In this forwarding method, each node receives sensing data packets from all its child nodes in different time slots and sends at once an aggregated data packet to its parent node in another time slot” (lines 164-166).

When each child node sends data to the parent node, it waits predefined period for the acknowledgement message that indicates whether data has been received correctly. If it receives acknowledgement message within this period, it realizes that data has been received successfully. Otherwise, it retransmits the sensing data. Enough length for a time slot allotted to each node is determined, so the time slot accommodates the above process.  

[Reviewer 2, Comment #4]

Additionally, it must be enumerated the requirements of using the proposal: it can be used in a unidirectional WSN where data only flows from sensor nodes to sink nodes (or parents)? 

[Author’s Response to Comment #4]

 Apologies for confusion. In this work, we considered only one directional data flow that from leaf nodes to the sink node. In Figure 8, we described the data collection process in our network model. As we mentioned in Response 3, parent nodes only acknowledge to child nodes that their data is received successfully. However, data is flowing from child nodes to the sink node. When we explained the example in Figure 8, we said,

In every active period, a node  wakes up and obtains its sensing data  of size S from its sensor. If  is a leaf node, it forwards Ds to its parent node. If  is a parent node, it receives a set of sensing data  from all child nodes . The parent node  then aggregates the set of sensing data with its own sensing data. Finally,  forwards the aggregated data to its parent node” (lines 439-443).   

In this article, we mentioned our target application in the lines 133 – 141. In this application we always collect the data from distributed sensor nodes. So, we did not consider the bidirectional data flow.

[Reviewer 2, Comment #5]

How a failed node can be detected by the others? As different WSN technologies and protocols exist, which of them are valid for using your proposal? 

[Author’s Response to Comment #5]

In the manuscript, we stated how faulty parent is detected by child nodes.

we identify as faulty nodes the parent nodes that do not respond (acknowledge) to their child nodes within a given time duration” (lines 125-126).

Our proposed technique is more appropriate to spanning tree structure based WSNs. Since we consider energy efficiency in recovery process, proposed method may perform better in Low Power TDMA MAC protocol based WSNs.

[Reviewer 2, Comment #6]

If a failed node is restored (by example because the use of energy harvesting techniques charges the battery some time later), how does the protocols respond?

[Author’s Response to Comment #6]

In this article, we consider the fault mostly due to the low battery power and target applications such as monitoring a temperature, pollution or toxic gas require large-scale network which may cover the entire city. We added additional details regarding the energy harvesting and battery replacement in the main manuscript.

[The following sentences has been added/modified to the manuscript]

On Page 4 Line 161

“In most environmental monitoring, infrastructure or facility diagnosis and wireless metering applications, employing energy harvesting devices like solar cell is not an effective solution since sensor devices are usually installed in indoor environment or dark basements of the buildings. Using a larger battery is not acceptable due to the cost and size constraints on the sensor devices, since these applications are often deployed throughout the entire city to monitor temperature, air pollution or toxic gas level”.   

However, restored parent may join the network after several cycles, which it listens to specify empty slot within the wireless range. If empty slots exist, it may use one of them to declare its existence. Then, original child nodes of this parent restore their initial connections. As we said, we did not consider restoring faulty parent in this article.  

[Reviewer 2, Comment #7]

About isolated nodes authors compares with a naïve recovery method (based on random selection) Why? Other alternatives?

[Author’s Response to Comment #7]

Thanks for your kind remark. We compared proposed method with other alternatives in Section 5, Figure 11(b). In this experiment, we compared our methods with reference DCFR algorithm. The line graph presented in Figure 5, is added without changing from authors’ previous work.

 [Reviewer 2, Comment #8]

Finally, a study of the data length and the time slot should be interesting. How it is defined the time-slot?  The relation between the time slot duration and the number of child’s and data length should be studied. What happens if child’s data length exceeds the slot duration and no more parents are available? How this situation is treated by the algorithms?

[Author’s Response to Comment #8]

As each node is assigned with fixed time-slot to execute transmission, parent nodes are allowed to adapt only predefined number of child nodes. In Section 4, we discussed about the relation of aggregated data length to the number of child nodes. In the given example Figure 8, we mentioned that the maximum allowed length of aggregated data is 8S. If parent node’s aggregated data is larger than 8S, it cannot complete transmission in given time-slot. Therefore, we use this constraint during the process of searching for the optimal NBP. The Eq. (9) is applied as constraint during the determination of NBP’s optimality. Following is brought from manuscript, On Page 14 Line 480

The length of each aggregated data must be shorter than the time slot constraint . The routing algorithm selects route paths that meet this constraint. The proposed algorithm NE-MCR also ensures that this constraint is satisfied when it searches for the optimum NBP.

     The constraint on the aggregated data length for NE-MCR is given by Eq. (9).

                                                                      (9)

Here,  is the length of ’s packet that is aggregated with the data received from all its child nodes.  is the length of ’s aggregated packet that is forwarded from  to ”.

In our implementation, if a child node cannot find the NBP that is able to accommodate the child nodes data within its slot, the child node informs the MCR_BP about unavailability of optimal NBP in its wireless range. During the implementation, we observed a few cases when all child nodes of MCR_BP were not able to find optimal NBP owning to this constraint. Hence, we are targeting this issue as our future work.   

Reviewer 3 Report

Thanks for inviting me to review the paper entitled 'An energy-efficient fail recovery routing in TDMA MAC protocol based wireless sensor network'. This paper proposes an energy efficient fail recovery routing method to operate over a data aggregation network topology. A fault recovery routing algorithm for TDMA-based wireless sensor networks to find an optimal neighbor backup parent in an energy efficient way. This study is interesting and useful for research and applications on wireless sensor network. Overall the paper is well written. I have some comments and questions for further improvement. 

(1) The Introduction is good, but it would be better if you clearly state what are others' work and what are your opinions, and please include references for others' work. A lot of statements seems subjective and there is no reference. 

(2) The paper is readable, but some sentences are not well written. For example, you do not need to say 'To the best of our knowledge', because everything in your paper should be 'To the best of our knowledge'. Also, a lot of abbreviations are used. Some of the abbreviations are not defined, compromising the readability. For example, what does CSMA stand for? In fact, many abbreviations are unnecessary. 

(3) This study is based on a lot of assumptions. Many of the assumptions look reasonable yet they are not justified by the authors. I suggest the authors add justifications or references for the assumptions. For instance, the assumptions in Line 199, 221, 225, 240, 288, 402, etc.

(4) Derivation process of the equations should be clearer. Please add references for the equations if you did not derive the equations by yourself in this paper. For example, how did you derive Eq. (7)?

(5) Figure 6 shows the algorithm. I expect to see more details about the implementation of the algorithm in resolving problem. A clearly stated example would be useful. The strength and constraints of the proposed method can be clarified through the example or case study. 

(6) The conclusion is more like a summary. Please rewrite the conclusions by specifying the key findings of this study. 

Author Response

Reviewer: 3

Thanks for inviting me to review the paper entitled 'An energy-efficient fail recovery routing in TDMA MAC protocol based wireless sensor network'. This paper proposes an energy efficient fail recovery routing method to operate over a data aggregation network topology. A fault recovery routing algorithm for TDMA-based wireless sensor networks to find an optimal neighbor backup parent in an energy efficient way. This study is interesting and useful for research and applications on wireless sensor network. Overall the paper is well written. I have some comments and questions for further improvement.

[Reviewer 3, Comment #1]

The Introduction is good, but it would be better if you clearly state what are others' work and what are your opinions, and please include references for others' work. A lot of statements seems subjective and there is no reference. 

[Author’s Response to Comment #1]

Apologies for lack references in the Introduction. We added reference to all statements we gave in the Introduction. The references are added in following lines: 31, 33, 35, 37, 40, 44, 45, 47, 50, 55, 56, 62, 66. Some new references are added in the reference list.   

[Reviewer 3, Comment #2]

The paper is readable, but some sentences are not well written. For example, you do not need to say, 'To the best of our knowledge', because everything in your paper should be 'To the best of our knowledge'. Also, a lot of abbreviations are used. Some of the abbreviations are not defined, compromising the readability. For example, what does CSMA stand for? In fact, many abbreviations are unnecessary.

[Author’s Response to Comment #2]

We appreciate your kind remarks. We changed “To the best of our knowledge” to “Authors believe that”. Definition for the CSMA is given in the introduction part of manuscript (line 36)

On Page 2 Line 45

In the past, many WSNs employed a carrier sense multiple access (CSMA) protocol due to its simplicity”.  

As you said, we used many abbreviations in current manuscript. The most abbreviations represent the algorithms introduced in related work. The frequently used abbreviations denote the procedures of proposed method such as MCR and NE-MCR. Those abbreviations, we referred many time to explain the different procedures and cases. Some of them only denotes special node identified through above procedures, e.g. MCR_BP. If we use always full name, it may create additional difficulties for readers.

[Reviewer 3, Comment #3]

This study is based on a lot of assumptions. Many of the assumptions look reasonable yet they are not justified by the authors. I suggest the authors add justifications or references for the assumptions. For instance, the assumptions in Line 199, 221, 225, 240, 288, 402, etc.

[Author’s Response to Comment #3]

We added some reference articles to support the assumptions made in current manuscript. Reference [27] added to support assumption in line 199. The work in [26] presents theoretical solution to achieve the assumption mentioned in line 221. The reference work in [27] supports the assumption made in line 225. We added reference [29] and [28] to support the assumptions claimed in line 240 and 288 respectively.

[Reviewer 3, Comment #4]

Derivation process of the equations should be clearer. Please add references for the equations if you did not derive the equations by yourself in this paper. For example, how did you derive Eq. (7)?

[Author’s Response to Comment #4]

Thanks for your kind remark. As you suggested, we added reference work that original presented transmission function in Eq. (7).

 [The following sentences has been added/modified to the manuscript]

On Page 8 Line 296

According to the radio model used in [25], data transmission usually depends on the distance and packet length, as expressed in Eq. (7).

                                                           (7)

[Reviewer 3, Comment #5]

Figure 6 shows the algorithm. I expect to see more details about the implementation of the algorithm in resolving problem. A clearly stated example would be useful. The strength and constraints of the proposed method can be clarified through the example or case study. 

[Author’s Response to Comment #5]

In the Section 4, we presented an example for selecting optimal NBP. In Figure.9, a subtree of the network is illustrated and details regarding the selection procedure is given as follow:

On Page 16 from Line 537 till 548.

For example, Figure. 9 illustrates a subtree of a network to depict how NE-MCR selects an optimal pair of  and  for a node . NE-MCR repeats this selection process for every node  as a processing to find a recovery route path for the case where  indeed fails during normal operation. In Figure. 9, the potential faulty node  is highlighted. In this subtree, none of its child nodes  ’s can reach their grandparent , and thus the MCR algorithm fails in finding a backup parent. Therefore, NE-MCR attempts to find an optimal pair ( , ) as follows. NE-MCR checks the potential of each  and adds  to the candidate set of , if  meets the constraint of Eq. (16). For each  of the candidate set, NE-MCR finds a set of  nodes that satisfy the constraint of Eq. (17), and adds a pair ( , ) to a set of candidate pairs. NE-MCR then calculates the cost function  of every candidate pair, and selects the pair ( , ) of the minimum  as the optimal recovery backup nodes.”  

[Reviewer 3, Comment #6]

The conclusion is more like a summary. Please rewrite the conclusions by specifying the key findings of this study. 

[Author’s Response to Comment #6]

We modified the conclusion and remove the most part of previous one. Changes in conclusion can be seen On Page 20 Line 648
